# Nationwide geospatial analysis of county racial and ethnic composition and public drinking water arsenic and uranium

Irene Martinez-Morata [1] ✉, Benjamin C. Bostick [2], Otakuye Conroy-Ben [3], Dustin T. Duncan[4], Miranda R. Jones[5], Maya Spaur[1], Kevin P. Patterson [1], Seth J. Prins[4], Ana Navas-Acien[1] & Anne E. Nigra [1]

There is no safe level of exposure to inorganic arsenic or uranium, yet recent studies identified sociodemographic and regional inequalities in concentrations of these frequently detected contaminants in public water systems across the US. We analyze the county-level association between racial/ethnic composition and public water arsenic and uranium concentrations from 2000–2011 using geospatial models. We find that higher proportions of Hispanic/Latino and American Indian/Alaskan Native residents are associated with significantly higher arsenic and uranium concentrations. These associations differ in magnitude and direction across regions; higher proportions of non-Hispanic Black residents are associated with higher arsenic and uranium in regions where concentrations of these contaminants are high. The findings from this nationwide geospatial analysis identifying racial/ethnic inequalities in arsenic and uranium concentrations in public drinking water across the US can advance environmental justice initiatives by informing regulatory action and financial and technical support to protect communities of color.

In some US populations, drinking water is a significant source of exposure to inorganic arsenic and uranium, which are major environmental exposures associated with cancer, cardiovascular disease and other adverse health outcomes[1–6]. Inorganic arsenic (a metalloid) is a known human carcinogen, is consistently ranked number one on the Centers for Disease Control and Prevention (CDC)/Agency for Toxic Substances and Disease Registry's (ATSDR) Substance Priority List[7], and is also associated with adverse birth outcomes, diabetes, metabolic disease, and cardiovascular disease, even at low- to moderate-levels of exposure common in the US population[1,3,4,8]. While alpha radiation from uranium decay is classified as carcinogenic, growing epidemiologic evidence also indicates that uranium exposure is associated with chronic kidney disease and is toxic to the respiratory, neurologic, and reproductive systems[2,6,9,10]. The US Environmental Protection Agency (EPA) regulates both arsenic and uranium in US public drinking water systems, including community water systems (CWSs) which serve over 90% of residents year-round[11]. Although the US EPA sets a maximum contaminant level (MCL) of 30 μg/L for uranium and 10 μg/L for arsenic, EPA's non-enforceable maximum contaminant level goal for both is 0 μg/L because there is no known safe level of exposure to either inorganic arsenic or uranium (Denmark and the states of New Jersey and New Hampshire set more health-protective arsenic MCLs of 5 μg/L, and drinking water providers in the Netherlands adopted a voluntary standard of 1 μg/L in 2016)[12–15].

Recent studies indicate that both metals are frequently detected in CWSs and significant geographic and sociodemographic inequalities in CWS concentrations exist across the US[16,17]. CWSs reporting the highest concentrations of arsenic and/or uranium were those serving

[1]Department of Environmental Health Sciences, Columbia University Mailman School of Public Health, New York, NY, USA. [2]Lamont-Doherty Earth Observatory of Columbia University, Palisades, NY, USA. [3]School of Sustainable Engineering and the Built Environment, Arizona State University, Tempe, AZ, USA. [4]Department of Epidemiology, Columbia University Mailman School of Public Health, New York, NY, USA. [5]Department of Epidemiology, Johns Hopkins Bloomberg School of Public Health, Baltimore, MD, USA. ✉e-mail: im2557@cumc.columbia.edu

communities characterized as Hispanic/Latino or American Indian, those located in the Central Midwest or Southwest, those serving incarcerated populations in the Southwest, and those reliant on groundwater[16–19]. Structural racism likely underlies these inequalities in drinking water quality; water contaminant exposure disparities are driven by a complex, interactive, multilevel system structured around the natural (e.g. hydrogeology, climate, soil), built (e.g. agriculture, water infrastructure, land-use patterns), and sociopolitical (e.g. regulatory policy, historical settlement patterns) environments[20–24]. These documented disparities in CWS-level arsenic and uranium across sociodemographic subgroups underscore the need for robust geospatial analyses of community-level racial/ethnic composition and drinking water exposures to account for spatial dependency in CWS concentrations, which can yield biased effect estimates in traditional regression models[25]. Previously, nationwide studies evaluating these associations were not possible because nationwide estimates of contaminant concentrations were not available (distribution boundaries are not publicly available for the majority of CWSs). However, our team recently developed county-level, population-weighted estimates of contaminant concentrations in CWSs across the US for the 2000–2011 time period, which were developed from routine compliance monitoring records (EPA requires CWSs to routinely monitor for regulated contaminants, including arsenic and uranium)[16,17].

In this study, we examined the county-level association between racial/ethnic composition (the proportion of residents belonging to a given racial/ethnic group) and CWS arsenic and uranium concentrations across the conterminous US using geospatial models. Our objectives were to (a) evaluate the county-level distribution and spatial autocorrelation of CWS arsenic and uranium, (b) assess the nationwide county-level association between racial/ethnic composition and CWS arsenic and uranium concentrations in geospatial models, and c) quantify the strength, direction, and spatial stationarity of these associations at the local level. We specifically assessed geometric mean ratios in CWS metal concentrations per 10% higher proportion of residents who were non-Hispanic Black, non-Hispanic White, Hispanic/Latino, and American Indian/Alaskan Native. We predicted that higher proportions of non-Hispanic White residents would be associated with lower CWS arsenic and uranium concentrations (likely reflecting white supremacy and structural racism related to public drinking water infrastructure, investment decisions, and regulatory action that specifically benefit non-Hispanic White communities, which have been previously documented for some water exposures)[23,24,26,27], and that higher proportions of Hispanic/Latino and American Indian residents would be associated with higher CWS arsenic and uranium concentrations (also consistent with previous documented inequalities in public drinking water infrastructure and regulatory action across population groups)[20,23,28,29]. We also anticipated that the association between racial/ethnic composition and CWS arsenic and uranium concentrations would be unique for each racial/ethnic group and would differ in magnitude and direction of the association across different regions of the US because of regional differences in racial/ethnic composition, geologic context, urban vs rural landscapes, and land-use patterns relevant for drinking water arsenic and uranium concentrations. As a secondary analysis, we also assessed the association between racial/ethnic composition and CWS selenium and barium because these metals/metalloids are also regulated in CWSs by the EPA, are frequently detected in CWSs, and previous studies found significant inequalities in concentrations across sociodemographic and regional groups in CWS-level analyses[16] (although the health impacts of chronic, low-level exposure to selenium and barium are less clear[5,30,31].

In this work, we show that higher proportions of Hispanic/Latino and American Indian/Alaskan Native residents are associated with significantly higher CWS arsenic and uranium concentrations at the county-level, while higher proportions of non-Hispanic White

residents are associated with lower CWS arsenic and uranium concentrations. The association between racial/ethnic composition and CWS arsenic and uranium concentrations differs in magnitude and direction across regions; higher proportions of non-Hispanic Black residents is associated with higher CWS arsenic and uranium in regions where concentrations of these contaminants are high. Findings from this study can advance environmental justice initiatives by informing federal and state-level infrastructure investments, financial and technical support for CWSs, and additional regulatory action to protect communities disproportionately exposed to major public drinking water contaminants.

## Results

### Spatial distribution and clustering of arsenic and uranium in CWSs

Our analysis uses previously developed, county-level, population-weighted concentration estimates of CWS arsenic (2006–2011, $N = 2585$ counties) and uranium (2000–2011, $N = 1174$ counties)[16,17]. Maps showing the mean concentration estimates of county-level CWS arsenic and uranium across the conterminous US are presented in Fig. 1. Both county-level CWS arsenic and uranium exhibited significant positive global spatial autocorrelation (Moran's $I = 0.62$, $p$ value < 0.001 for arsenic, and $I = 0.68$, $p$ value < 0.001 for uranium). We next identified specific areas of the US where spatial autocorrelation was present by assessing Local Indicators of Spatial Association (LISA) clusters. We observed significant High-High and Low-Low LISA clusters for both arsenic and uranium, which are presented in Fig. 1. Areas with High-High significant spatial autocorrelation for both arsenic and uranium were located in the Midwest and Southwest (e.g. Nevada, California, Arizona, New Mexico, Texas, Kansas, Nebraska) where CWS arsenic and uranium concentrations are highest in underlying groundwater[16,17,32]. Low-Low clusters for arsenic were identified in the Southeast and Mid-Atlantic regions (e.g. Alabama, Arkansas, North and South Carolina, Virginia, Kentucky). Low-Low clusters for uranium were located in similar geographic areas, although spatial coverage was poorer in the eastern US. We also identified significant positive spatial autocorrelation for barium and selenium (Supplementary Fig. 1).

### County-level sociodemographic characteristics and metal concentrations

Because we restricted our analysis of each racial/ethnic group to counties with at least 100 residents of that group to avoid positivity violations[33] (i.e., the number of residents in a given racial/ethnic subgroup should be greater than zero for all counties analyzed or effect estimates would be extrapolated beyond the range of actual values), the number of counties included in the analysis of each racial/ethnic group differed (see Methods setion: Exclusion criteria). Across the counties included in each racial/ethnic analysis and all counties included in any analysis ($N = 2653$), we first described mean county-level CWS arsenic and uranium concentrations, racial/ethnic composition, and other county-level characteristics of interest (Table 1). Nationwide, the county-level mean percentage of residents belonging to each racial/ethnic group was 7% (non-Hispanic Black), 2% (American Indian/Alaskan Native), 9% (Hispanic/Latino), and 80% (non-Hispanic White). In analyses for each racial/ethnic group, we evaluated a total of 1881 (non-Hispanic Black), 1575 (American Indian/Alaskan Native), 2401 (Hispanic/Latino), and 2653 (non-Hispanic White) counties. Compared to counties included in any analysis, counties included in the analysis for non-Hispanic Black residents had lower mean CWS arsenic (1.17 µg/L vs. 1.42 µg/L) and uranium (2.5 µg/L vs. 3.45 µg/L) concentrations, a lower percentage of CWSs reliant on groundwater, and a lower percentage of the population living in rural areas. Counties included in the analysis for American Indian/Alaskan Native residents had larger population size and density and the lowest percentage of the population living in rural areas. Counties included in the analysis

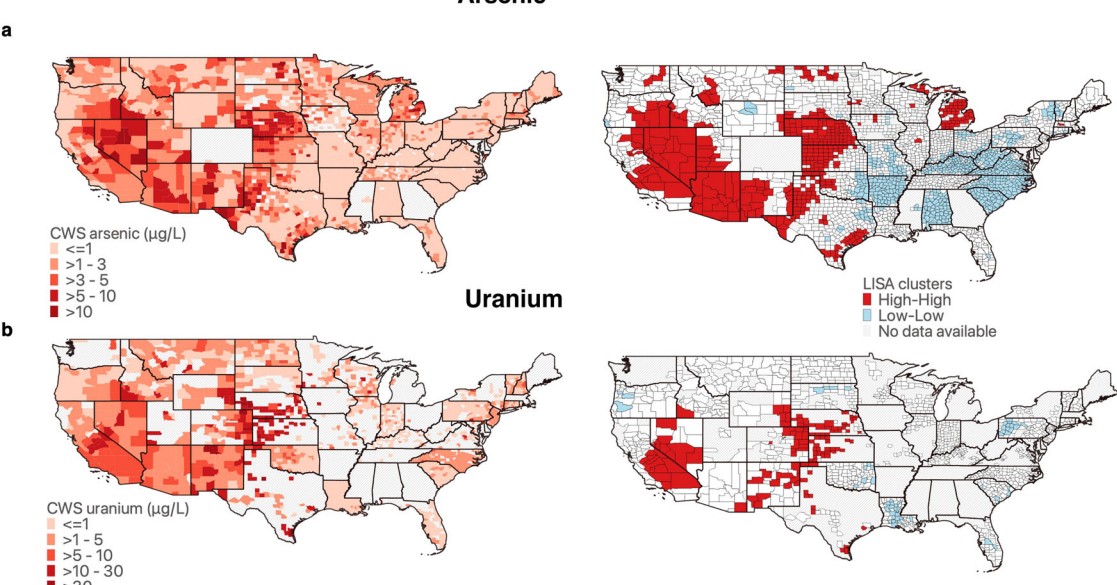

**Fig. 1 | Nationwide spatial distribution of county-level community water system (CWS) metal concentrations and local indicators of spatial associations (LISA) clusters. a** Arsenic. **b** Uranium. County-level CWS metal concentrations were originally developed and described by Nigra et al. (2020) and Ravalli et al. (2022) and are categorized and colored in red scale. The highest concentration categories correspond to the current MCL (10 μ/L for arsenic and 30 μ/L for uranium). High-High LISA clusters are shown in red, and Low-Low LISA clusters are shown in blue. Counties overlaid with gray hatch marks are missing CWS metal concentration estimates. All counties included in the analyses had at least 1 neighbor county.

**Table 1 | County-level mean estimated community water system (CWS) metal concentrations and sociodemographic characteristics for all counties included in any analysis (N = 2631 counties), and separately for counties included in analyses specific to each racial/ethnic group (counties with >100 residents of each racial/ethnic group)**

|  | All counties in any analysis | non-Hispanic Black[a] | American Indian/ Alaskan Native[a] | Hispanic/Latino[a] | non-Hispanic White[a] |
|---|---|---|---|---|---|
| *N* | 2631[b] | 1881 | 1575 | 2401 | 2631 |
| *CWS metal estimates (mean, SD)* |  |  |  |  |  |
| Arsenic (2006–2011)[c] | 1.42 (2.22) | 1.17 (1.85) | 1.37 (1.98) | 1.36 (2.05) | 1.42 (2.22) |
| Uranium (2000–2011)[c] | 3.45 (6.85) | 2.5 (4.72) | 2.73 (4.98) | 3.38 (6.71) | 3.45 (6.85) |
| *Sociodemographic characteristics* |  |  |  |  |  |
| Population size (mean, SD) | 103,318 (330,535) | 141,333 (386,162) | 163,453 (418,407) | 113,554 (345,861) | 103,318 (330,535) |
| Population density (mean, SD)[d] | 180 (600) | 247 (702) | 270 (756) | 194 (628) | 180 (600) |
| % public drinking water sourced from groundwater supplies (mean, SD)[e] | 62 (42) | 56 (43) | 56 (42) | 61 (42) | 62 (42) |
| Median household income (mean, SD) | 44,072 (10,611) | 44,664 (11,353) | 46,147 (11,479) | 44,514 (10,782) | 44,072 (10,611) |
| % adults with high school diploma (mean, SD) | 83 (9) | 82 (9) | 82 (9) | 83 (9) | 83 (9) |
| % population living in rural area (mean, SD) | 58 (31) | 48 (28) | 45 (28) | 55 (30) | 58 (31) |
| *Racial/ethnic composition (mean, SE)* |  |  |  |  |  |
| % non-Hispanic Black | 7.1 (12.2) | 9.9 (13.6) | 7.5 (11) | 7.5 (12.1) | 7.1 (12.2) |
| % American Indian/Alaskan Native | 2.1 (7.5) | 1.36 (4.1) | 3.26 (9.6) | 1.9 (6.7) | 2.1 (7.5) |
| % Hispanic/Latino | 8.7 (13.9) | 9.2 (13.5) | 10.1 (13.7) | 9.5 (14.4) | 8.7 (13.9) |
| % non-Hispanic White | 79.6 (19.1) | 76.7 (18.5) | 76 (18.5) | 78.6 (18.9) | 79.6 (19.1) |

[a]*N* represents the total number of conterminous US counties evaluated for each racial/ethnic group after restricting to counties with >100 residents of the racial/ethnic group of interest.
[b]*N* = 2585 is the total number of conterminous US counties with data available for at least one CWS metal exposure estimate, the percent of public drinking water sourced from groundwater supplies, population density, median household income, and the percent of adults with a high school diploma.
[c]Of the total 2631 conterminous US counties, *N* = 2585 had arsenic CWS exposure estimates available and *N* = 1174 had uranium CWS exposure estimates available. Note that uranium CWS exposure estimates are aggregated to the 2000–2011 averaging period to account for differences in compliance monitoring requirements associated with the US EPA's Radionuclides Rule.
[d] Population density is calculated as number of residents per square mile.
[e]The percent of public drinking water sourced from groundwater supplies was calculated using nationwide estimates of water use published by the US Geological Survey for 2010.

for Hispanic/Latino residents had higher mean CWS uranium (3.38 μg/L vs. 3.45 μg/L) concentrations. No counties were excluded for having <100 residents in the analysis for non-Hispanic White residents, and therefore these counties are the same as all counties included in any analysis.

**Spatial lag regression**
To identify nationwide inequalities in CWS arsenic and uranium by racial/ethnic composition, we quantified the association between county-level racial/ethnic composition and CWS arsenic and uranium separately for each of the four racial/ethnic groups using spatial lag

**Table 2 | Geometric mean ratios (95% CI) of county-level community water system (CWS) arsenic and uranium concentrations per a 10 percent higher proportion of non-Hispanic Black, American Indian/Alaskan Native, Hispanic/Latino, and non-Hispanic White residents, from spatial lag regression models**

| | % Non-Hispanic Black | | % American Indian/Alaskan Native[a] | | % Hispanic/Latino | | % Non-Hispanic White | |
|---|---|---|---|---|---|---|---|---|
| | GMR | % Change | GMR | % Change | GMR | % Change | GMR | % Change |
| *Arsenic* | | | | | | | | |
| N | 1848 | | 1522 | | 2341 | | 2585 | |
| Model 1 | 0.92 (0.90, 0.94) | −8% (−10, −6) | 1.07 (1.03, 1.11) | 7% (3, 11) | 1.06 (1.04, 1.08) | 6% (4, 8) | 0.99 (0.98, 1.00) | −1% (−2, 0) |
| Model 2 | 0.93 (0.91, 0.95) | −7% (−9, −5) | 1.06 (1.02, 1.10) | 6% (2, 10) | 1.06 (1.04, 1.08) | 6% (4, 8) | – | – |
| *Uranium* | | | | | | | | |
| N | 832 | | 778 | | 1070 | | 1174 | |
| Model 1 | 0.91 (0.87, 0.95) | −9% (−13, −5) | 1.02 (0.96, 1.08) | 2% (−4, 8) | 1.17 (1.13, 1.22) | 17% (13, 22) | 0.95 (0.92, 0.97) | −5% (−8, −3) |
| Model 2 | 0.97 (0.92, 1.01) | −3% (−8, 1) | 1.05 (0.99, 1.12) | 5% (−1, 12) | 1.17 (1.13, 1.22) | 17% (13, 22) | – | – |

Spatial autocorrelation was modeled in Lagrange models with autoregressive correlation structure. County-level CWS arsenic and uranium were natural log-transformed for analysis. Model 1 adjusts for population density, the percent of public water sources from groundwater supplies, median household income and the percent of adults with a high school diploma. Model 2 further adjusts for the racial/ethnic composition of other racial/ethnic groups, except non-Hispanic White (leave-one-out model). Coefficients for the spatial lag term were significant in all models.
[a]Sensitivity analyses not adjusting for population density yielded similar findings.

regression models with an autoregressive correlation structure to account for spatial dependence (see Methods section: Statistical analysis: spatial lag regression)[34]. We evaluated the geometric mean ratios (GMRs) and corresponding percent differences in CWS arsenic and uranium concentrations per a 10 percent higher proportion of residents belonging to each racial/ethnic group in progressively adjusted models (Table 2).

Our main model of interest (Model 1) adjusted for population density, the percent of public drinking water sourced from groundwater supplies, median household income, and the percent of adults with a high school diploma. Effect estimates were considered statistically significant when 95% confidence intervals did not cross the null (GMR of 1 and percent change of 0). In fully adjusted models (Model 1), the percent increase in the geometric mean (95% CI) of CWS metal concentrations per 10% higher proportion of Hispanic/Latino residents was 6% (4, 8) for arsenic and 17% (13, 22) for uranium. For a 10% higher proportion of American Indian/Alaskan Native residents, the percent increase in the geometric mean (95% CI) of CWS metal concentration was 7% (3, 11) for arsenic and 2% (−4, 8) for uranium. For a 10% higher proportion of non-Hispanic Black residents, the percent decrease in the geometric mean (95% CI) of CWS metal concentration was 8% (10, 6) for arsenic and 9% (13, 5) for uranium. The percent decrease in the geometric mean (95% CI) of CWS metal concentrations for a 10% higher proportion of non-Hispanic White residents was 1% (2, 0) for arsenic and 5% (8, 3) for uranium. Models further adjusting for the composition of other racial/ethnic groups yielded similar results (Model 2).

We performed several sensitivity analyses, all with similar findings. Adjusting for the CDC/ATSDR's county-level index of socioeconomic status[35] (rather than median household income and the percent of adults with a high school education) yielded similar findings, supporting that the adjustment for income and education in Model 1 (derived from our conceptual framework in Supplementary Fig. 1) was able to capture socioeconomic status. We conducted several additional sensitivity analyses that also yielded robust findings, including: further adjustment for the total number of non-Hispanic White residents, using county-level 95th percentile CWS arsenic and uranium concentrations (rather than the mean), assessing higher proportions of residents in a given racial/ethnic group corresponding to the interquartile range and 60% (rather than 10%), without adjustment for population density, and after restriction to counties in the western US where CWS arsenic and uranium are highest (see Methods: Statistical analysis: spatial lag regression and Supplementary Table 1). As a complementary analysis to our assessment of higher proportions

of non-Hispanic White residents, we also assessed the association per a 10% higher proportion of all residents who were not categorized as non-Hispanic White, including non-Hispanic Black, American Indian/Alaskan Native, Hispanic/Latino, non-Hispanic Asian, Native Hawaiian or Other Pacific Islander, and all residents in other racial/ethnic categories. For a 10% higher proportion of these residents (all residents who were not categorized as non-Hispanic White), the percent increase in the geometric mean (95% CI) was 1% (−1, 3) for arsenic and 10% (6, 14) for uranium (results are shown in Supplementary Table 2). Because this association persisted even after combining all residents not categorized as non-Hispanic White, these complementary findings support the general hypothesis and leading theoretical framework on water exposure disparities- that white supremacy and structural racism are associated with drinking water inequalities[23]. However, the specific mechanisms underlying these disparities (e.g., infrastructure investments or management decisions specifically benefiting non-Hispanic White communities, etc.) are beyond the scope of this analysis. Findings were also similar for barium and selenium (Supplementary Table 3).

**Geographically weighted regression**

To assess local inequalities in CWS arsenic and uranium by racial/ethnic composition we next explored the strength and direction of the association between racial/ethnic composition and CWS arsenic and uranium, and evaluated spatial non-stationarity in the effect estimates via geographically weighted regression models (see Methods section: Statistical analysis: Geographically weighted regression). County-specific coefficients from the geographically weighted regression analysis for each racial/ethnic group and CWS arsenic and uranium are displayed in Figs. 2 and 3, respectively. County colors represent the magnitude of the local GMR of CWS metals for a 10% higher proportion of residents in each specified racial/ethnic group, with positive associations (GMR > 1) displayed in red and inverse (GMR < 1) associations displayed in blue. Positive associations imply that the CWS metal concentrations are higher in the county than predicted from the regression model, potentially due to local-scale heterogeneity.

We observed spatial non-stationarity in the associations between racial/ethnic composition and CWS metal concentrations for all racial/ethnic groups for both arsenic and uranium, with different associations at the local level for each racial/ethnic group. Although the GMRs of CWS metal concentrations per a 10 percent higher proportion of non-Hispanic Black residents were inverse in nationwide spatial lag regression models (Table 2), geographically weighted regression indicated a positive association for CWS arsenic and uranium in areas

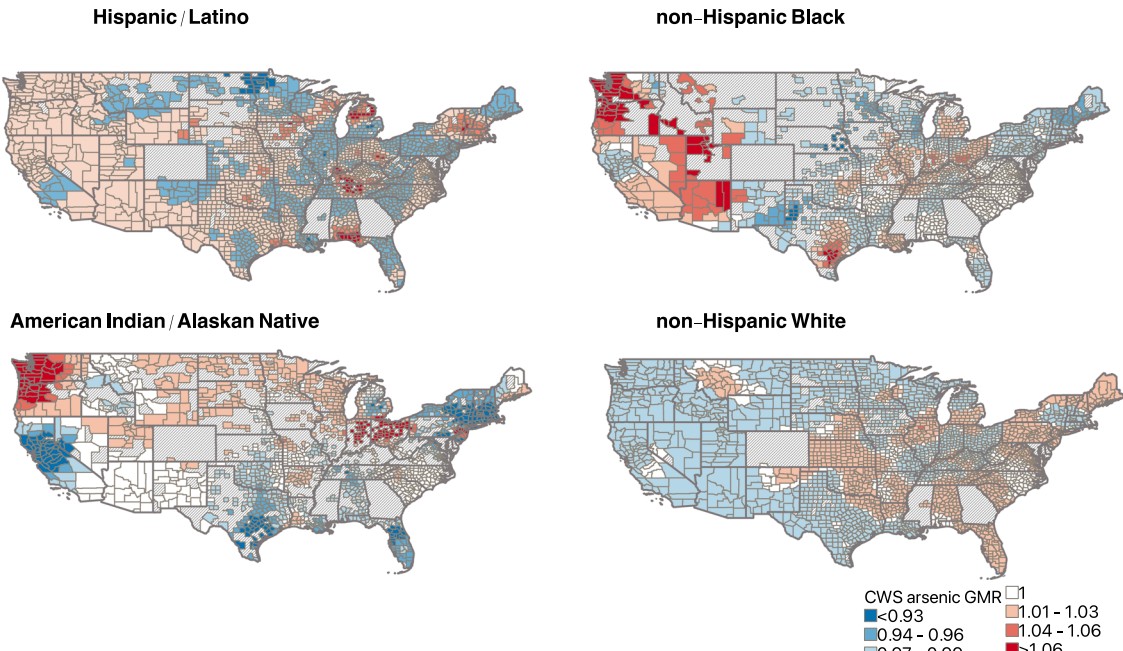

**Fig. 2 | Direction and magnitude of the geometric mean ratio (GMR) of county-level community water system (CWS) arsenic concentrations per 10 percent higher county proportion of non-Hispanic Black, American Indian/Alaskan Native, Hispanic/Latino, and non-Hispanic White residents from geographically weighted regression models.** Models were adjusted for population density, the percent of public drinking water supplied from groundwater sources, median household income, and the percent of residents with a high school diploma. Counties with a positive effect estimate (GMR > 1) are shown in red scale colors, counties with a negative effect estimate (GMR < 1) are shown in blue scale colors, and counties where a null effect estimate was observed are shown in white. Counties with missing data are shown with light gray hatch marks. The number of counties included in the geographically weighted regression analysis for each racial/ethnic group were: non-Hispanic Black ($n = 1848$), American Indian/Alaskan Native ($n = 1522$), Hispanic/Latino ($n = 2341$), and non-Hispanic White ($n = 2585$). Photocopy-friendly versions of these maps (contrasts visible in black and white) are available in Supplementary Fig. 8.

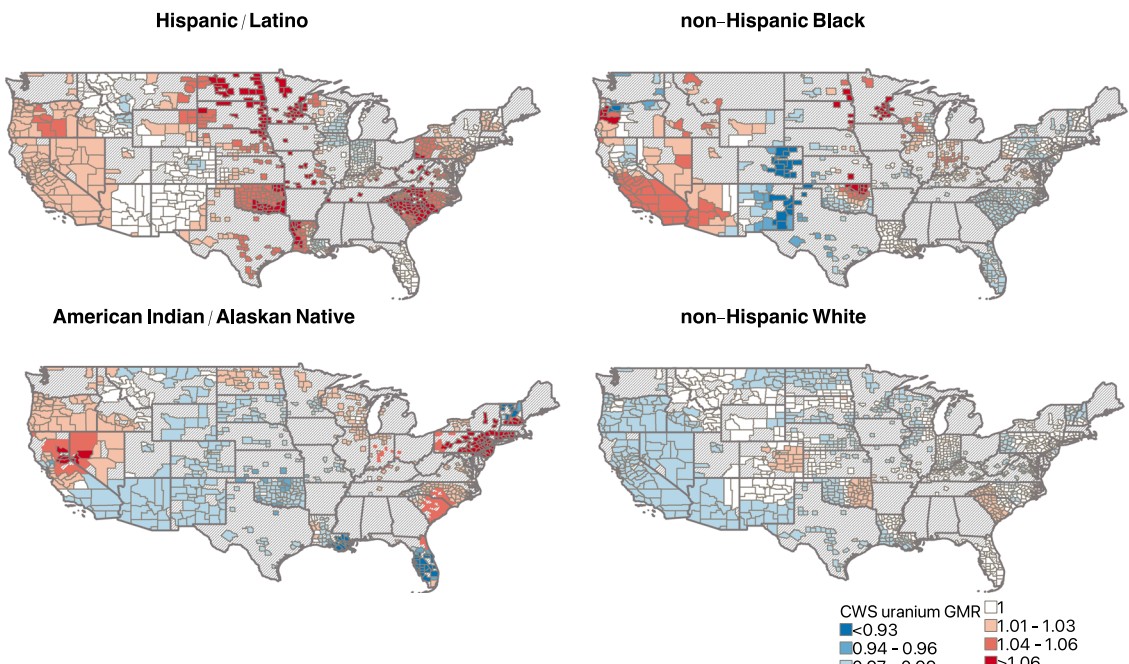

**Fig. 3 | Direction and magnitude of the geometric mean ratio (GMR) of county-level community water system (CWS) uranium concentrations per 10 percent higher county proportion of non-Hispanic Black, American Indian/Alaskan Native, Hispanic/Latino, and non-Hispanic White residents from geographically weighted regression models.** Models were adjusted for population density, the percent of public drinking water supplied from groundwater sources, median household income, and the percent of residents with a high school diploma. Counties with a positive effect estimate (GMR > 1) are shown in red scale colors, counties with a negative effect estimate (GMR < 1) are shown in blue scale colors, and counties where a null effect estimate was observed are shown in white. Counties with missing data are shown with light gray hatch marks. The number of counties included in the geographically weighted regression analysis for each racial/ethnic group were: non-Hispanic Black ($n = 832$), American Indian/Alaskan Native ($n = 778$), Hispanic/Latino ($n = 1170$), and non-Hispanic White ($n = 1174$). Photocopy-friendly versions of these maps (contrasts visible in black and white) are available in Supplementary Fig. 8.

where these CWS concentrations are high (particularly in the southwest, see Fig. 1). This was particularly evident for arsenic in western Texas, Arizona, California, Nevada, Utah, Oregon, and Washington, and for uranium in Arizona, California, Nevada, Oregon, and Oklahoma where we observed positive associations of higher magnitude (Figs. 2 and 3). Similarly, we observed a positive association between higher proportions of American Indian/Alaskan Native residents and CWS arsenic and uranium in the Northern Plains and in the west, particularly across Oregon, Washington, Montana, North Dakota and South Dakota, Minnesota, and Wisconsin for arsenic, and California, Nevada, Oregon, North Dakota, and Wisconsin for uranium. Findings for Hispanic/Latino residents were more consistent with findings from global spatial lag models, with significant, positive associations between higher proportions of Hispanic/Latino residents and CWS arsenic and uranium across most of the conterminous US. As anticipated, we observed higher model goodness-of-fit in areas with higher CWS metal concentrations and better spatial coverage of CWS arsenic and uranium estimates (Supplementary Figs. 3 and 4). Associations and goodness-of-fit for selenium and barium also differed locally and are presented in Supplementary Figs. 5 and 6.

## Discussion

Our study provides new evidence that racial/ethnic composition is associated with public drinking water arsenic and uranium concentrations, adding to a growing body of literature characterizing nationwide environmental and racial injustices in US public drinking water contaminants[16–20,26–29]. Although our analysis was informed by a strong conceptual framework (see Supplementary Fig. 1) and other previously well-developed frameworks describing how disparities in drinking water exposures result from structural and environmental racism[20–24], our study is descriptive in nature and is not intended to identify mechanisms by which racial/ethnic public water concentration disparities were created or reinforced.

As predicted, nationwide analyses indicated that a higher proportion of non-Hispanic White residents was associated with lower CWS arsenic and uranium, while a higher proportion of Hispanic/Latino and American Indian/Alaskan Native residents was associated with higher CWS arsenic and uranium. These findings remained robust across several sensitivity analyses considering alternative adjustment approaches and alternative magnitudes of change in racial/ethnic composition, and are also consistent with previous studies utilizing non-geospatial approaches at the water system level[20,26,29]. These findings likely reflect structural environmental racism and white supremacy and privilege.

Structural racism refers to the "totality of ways in which societies foster racial discrimination, through mutually reinforcing inequitable systems (in housing, education, employment, earnings, benefits, credit, media, health care, criminal justice, and so on)"[36,37]. Structural racism serves to maintain white supremacy, in part, by amassing resources for and benefiting white communities to the detriment of communities of color[38–40]. Extensive research in the fields of sociology and epidemiology support that racism is a fundamental cause of disease and producer of health inequalities across racial/ethnic groups[41–43]. Structural racism produces health inequalities for marginalized communities through numerous mechanisms, including through the creation and perpetuation of inequalities in toxic environmental exposures, social determinants of health, and other psychosocial stressors[36,37,39,44–46]. Prior studies identifying the specific mechanisms underlying water inequalities are abundant, although responsible mechanisms differ across contaminants, geographic regions, and the sociodemographic characteristics of impacted communities. Briefly, previous studies have identified community linguistic isolation, selective enforcement of drinking water regulations, the underbounding of communities of color from municipal boundaries and water services, and the direct withholding of resources and infrastructure investments as examples of mechanisms responsible for producing and maintaining inequalities in exposure to regulated drinking water contaminants[23,26,47–49]. For example, inequalities in public water lead exposure are related to changes in water supply sources, corrosion control, service line infrastructure investments, and targeted tap sampling for compliance monitoring[32,50–53]. Similar mechanisms may underlie the unequal distribution of arsenic and uranium concentrations in CWS observed in this study, although additional research is needed to evaluate those mechanisms in detail.

Consistent with our other prediction that these associations would differ across geographic regions due to underlying geologic context, (i.e., water metal concentrations are influenced by natural composition of rocks and soil, and anthropogenic contamination from industrial activity, mining, agriculture)[2,3,5] we found larger effect estimates from geographically weighted regression models in the southwestern US where water arsenic and uranium are high, and smaller effect estimates in other regions where these concentrations of these contaminants are low (i.e., the southeast). Taken together, these findings also suggest that communities with higher proportions of Hispanic/Latino and American Indian/Alaskan Native residents may be served by public water systems with elevated concentrations of other contaminants that were not evaluated in the present study, especially for contaminants that are present at meaningful concentrations in the geographic area.

There are several potential explanations for the observed inverse relationship between the proportion of non-Hispanic Black residents and CWS arsenic and uranium concentrations in nationwide analyses. First, nationwide, non-Hispanic Black residents are more likely to live in urban areas[54,55], and therefore may be more likely served by large urban CWSs which have higher economies of scale and can implement more health protective water treatment systems and decisions. Cities with higher proportions of non-Hispanic Black residents are also more likely located in the eastern and southeastern US, where CWS arsenic concentrations are low. Although the majority of Hispanic/Latino residents also live in urban areas, these residents are also more likely to live in the southwestern US where groundwater arsenic and uranium concentrations are naturally high[19,56,57]. This explanation is supported by our findings from geographically weighted regression (Figs. 2 and 3) and the analyses restricted to western counties (Supplementary Table 1), which indicated that a higher proportion of non-Hispanic Black residents was associated with higher CWS arsenic and uranium in the southwest, but not in the southeast where arsenic and uranium water concentrations are low (Fig. 1). These findings suggest that racial/ethnic inequalities in public water contaminant exposures may be more likely or more severe in geographic regions with both a high percentage of public water users of the specified racial/ethnic group and relatively high concentrations of specific contaminants in source water. Additional public water contaminants beyond metals should be also assessed, as these will likely display different nationwide geographic patterns which may overlap with areas where a majority of residents are non-Hispanic Black[23,24]. Future studies should assess these differences in analyses stratified by urban/rural areas to evaluate these potential explanations. Additionally, in areas of the US where non-Hispanic Black and American Indian communities rely heavily on private wells (the rural southeast and the rural Great Plains and southwest, respectively), public water systems may predominantly serve non-Hispanic White residents. Therefore, it is possible that the current analysis may underestimate the impact of community racial/ethnic composition on public water arsenic and uranium for both non-Hispanic Black and American Indian communities in these particular areas.

Across all analyses, our findings were largely consistent for arsenic and uranium, likely reflecting the clustering and co-occurrence of arsenic and uranium in public water systems which has been previously described in detail[16]. The co-occurrence of arsenic and uranium

has been documented in both regulated and unregulated water supplies[16,19,32,58,59], and may be related to carbonate complexes increasing the solubility of arsenic and uranium, and/or the presence of bicarbonate ions in oxic water sources (bicarbonate ions may be especially relevant for the continued co-occurrence of arsenic and uranium in treated water)[60–62].

Our findings are particularly relevant to public health because there is no safe level of exposure to inorganic arsenic and uranium[2,3]. At moderate to high levels of exposure, inorganic arsenic is associated with a number of adverse health outcomes including diabetes, lung disease, neurologic effects, kidney disease, and birth outcomes including low birth weight and impaired neurodevelopment; inorganic arsenic is also associated with cardiovascular disease and increased oxidative stress even at low to moderate exposure levels common in the US population[1,5,63–66]. Uranium exposure is associated with kidney disease, including kidney cancer, and neurologic and cardiovascular disease, although evidence for other adverse health outcomes is more limited[2,6,9,10,67]. Although the effect estimates described in the current study are small to modest, their public health implications may be substantial when considered across the entire population. Several studies have identified disparities in health outcomes across racial/ ethnic groups that are related to uranium and arsenic exposure. For example, American Indian/Alaskan Native populations experience substantially higher rates of kidney cancer compared to non-Hispanic White populations, especially in the Southwestern US where uranium exposures and co-exposures to arsenic and other metal mixtures are relatively high as a result of the uranium mining activity that occurred over the past century[19,58,59,68]. For example, the mean uranium concentration in urine (biomarker of internal dose) among pregnant women living in the American Indian Reservations in the Southwest is 2.67–2.8 times higher compared to the general US population[69].

Our study has several limitations. First, our nationwide evaluation of the county-level association between racial/ethnic composition and CWS metal concentrations may mask associations operating at finer geographic resolutions (e.g. within states, or across Census blocks or zip-codes)[21,70]. We were not able to evaluate these associations at finer geographic resolutions because nationwide CWS concentration estimates are not yet available at these levels of aggregation (there is currently no publicly available, nationwide map of public drinking water system distribution boundaries that can be used to generate concentration estimates at these resolutions). Future studies at smaller spatial scales and with finer geographic resolution may also address the substantial differences in county size across the US. Larger county sizes may lead to larger measurement error, which may especially impact findings in the western US compared to the eastern US where counties areas are smaller. One recent study at the Census-block level in California also found significant associations between higher proportions of residents of color and drinking water arsenic concentrations, suggesting that these associations are likely also present for other states in the southwestern US[71]. Although some studies have evaluated sociodemographic characteristics of the "city-served" by CWSs, which is reported in the US EPA's Safe Drinking Water Information System (SDWIS), approximately half of all public water systems did not report "city-served" in SDWIS[24,72]. In future studies, available water system distribution maps can be used to determine if "city-served" as reported in SDWIS provides a reasonable approximation of water system distribution boundaries nationwide.

Second, although our nationwide, county-level analysis was intended to minimize selection bias, we restricted our analyses to counties with at least 100 residents in the racial/ethnic group of interest to avoid positivity violations, and this may have introduced selection bias. Counties excluded from analysis under this criterion had smaller population sizes, lower population densities, a higher percentage of residents living in rural areas, higher proportions of non-Hispanic White residents, and higher CWS arsenic and uranium

concentrations (Supplementary Table 4). We did not assess associations for Asian and Native Hawaiian or Pacific Islander groups because no counties had >60% of residents for these groups. Future analyses can assess these associations at finer geographic resolutions, restricted to areas with larger populations of interest. We relied on Census bridged race categorized data, which does not capture several domains of racial/ethnic composition and could lead to misclassification. In addition, our analyses for uranium were limited by the relatively poor spatial coverage in CWS uranium concentration estimates, especially for the eastern and southeastern US. The CWS uranium concentrations used here are based on the most recently available CWS uranium monitoring records made publicly available by EPA, and are to our knowledge the most complete estimates currently available. Future studies can improve upon these estimates when EPA releases additional compliance monitoring records for the Fourth Six Year Review (expected 2023).

Future studies should also evaluate the association between racial/ethnic segregation and inequalities in CWS metal concentration estimates, especially because prior analyses of noise exposure[70] and air pollution[21,73,74] indicate that racial/ethnic segregation is associated with exposure inequalities. Geographically weighted regression indicated a positive association between larger proportions of non-Hispanic Black residents and CWS arsenic in parts of west Texas and the Midwest. These findings suggest that additional analyses at a finer geographic resolution should focus specifically on these particularly vulnerable geographic areas where metals in CWSs and soil are particularly high (public water system distribution boundaries are also publicly available for many of these states). Future analyses should also assess whether changes in racial/ethnic composition and migration patterns over time are associated with changes in CWS arsenic and uranium concentrations over time.

Current US public drinking water infrastructure, management, and regulatory action does not adequately protect communities of color from elevated contaminant exposures. Recent federal infrastructure investment proposals have suggested targeting financial and technical support to small public water systems in rural and tribal areas, which experience substantial difficulties in achieving lower contaminant concentrations[75]. Findings from our current study add to a mounting body of evidence supporting that financial and technical support, infrastructure investments, and regulatory action must be directed towards supporting communities with larger populations of American Indian/Alaskan Native, Hispanic/Latino, and non-Hispanic Black residents. Given the known inequalities in public drinking water lead exposure for non-Hispanic Black communities across the US, specific and additional financial and technical support should also be directed to public water systems serving these communities to reduce elevated water concentrations and associated health outcomes. Racial/ ethnic inequalities in public drinking water arsenic and uranium concentrations are unacceptable, and further research is needed to identify the specific mechanisms by which structural racism is operating to result in concentration inequalities for multiple regulated drinking water contaminants.

## Methods
### County-level public drinking water metal concentrations
We utilized previously developed, county-level, population-weighted concentration estimates of arsenic, uranium, selenium, and barium concentrations in CWSs across the US, which were previously developed and described in detail[17]. These concentration estimates are based on the most recent publicly available nationwide monitoring data for public water systems. Briefly, county-level CWS metal concentration estimates were developed using routine compliance monitoring records compiled by the US Environmental Protection Agency's (EPA) for the National Contaminant Occurrence Database supporting the Second and Third Six Year Review (SYR)[76]. The SYR

database contains routine compliance monitoring records voluntarily submitted to EPA by primacy agencies including states and tribal authorities[77,78]. CWS metal concentrations were averaged within a six-year period (2006–2011) for arsenic, selenium, and barium and within an eleven-year period (2000–2011) for uranium. These three- and eleven-year averages yield unbiased concentration estimates by accounting for the different compliance monitoring periods defined by EPA's Standardized Monitoring Framework requirements for arsenic (regulated under the Chemical Contaminants Rule and the Final Arsenic Rule) and for uranium (regulated under the Radionuclides Rule)[17]. Average metal concentrations at the county-level were calculated by weighing the average concentration for each CWS within the county by the size of the population served by each CWS. County-level CWS metal averages are missing when a) no compliance monitoring records were submitted for that county to the SYR, or b) CWSs serving that county reported serving <50% of the public-water reliant population in the entire county[17]. All county-level CWS metal variables were right skewed and natural log-transformed for analysis.

## County-level racial/ethnic composition and sociodemographic data

County-level racial/ethnic composition variables were derived from 2010 US Census Population Estimates and included the proportion of residents who are American Indian or Alaskan Native (hereafter referred to as American Indian/Alaskan Native), non-Hispanic Asian, Native Hawaiian or Other Pacific Islander, Hispanic or Latino (hereafter referred to as Hispanic/Latino), non-Hispanic Black or African American (hereafter referred to as non-Hispanic Black), and non-Hispanic White. We selected sociodemographic variables from time periods that overlapped with the time period of the CWS arsenic and uranium concentration estimates (2006–2011). The percent of adults with a high school diploma was derived from the 2007–2011 US Census American Community Survey. Median household income was derived from the 2011 Small Area Income and Poverty Estimates and 2010–2011 National Center for Education Statistics data. We also downloaded the CDC/ATSDR's 2014 county-level index for socioeconomic status, which we adjusted for in sensitivity analyses rather than adjusting for median household income and the percent of adults with a high school diploma (the CDC's county-level index for socioeconomic status is a rank-order index for all counties in the US based on the percent of adults living below the federal poverty line, the percent of unemployed adults, median household income, and the percent of adults without a high school diploma)[35]. Although other variables which reflect socioeconomic status were available and are likely related to both racial/ethnic composition and CWS metal concentrations (such as the percent of the population living in rural areas, the percent of residents without health insurance, percent unemployment, percent of children eligible for free lunch, and percent of children living below poverty level), these variables did not open biasing paths on our conceptual diagram (Supplementary Fig. 1), and we therefore did not adjust for them. County total population and population density (population per square mile) were also retrieved from the 2010 US Census. We estimated the percent of public drinking water supplied by groundwater sources (versus surface water sources) from estimates of total groundwater and surface water withdrawn for public drinking water calculated by the US Geological Survey for 2010 (we used 2015 estimates for 20 counties which were missing estimates in 2010; the Spearman correlation coefficient between estimates for the two years was 0.93)[56,57]. A full description of all county-level variables used in this analysis is available in Supplementary Table 5. All data management and analysis were conducted in R version 4.0.3, except the assessment of spatial autocorrelation which was conducted in GeoDa version 1.18.0. For visualization, maps were generated in QGIS version 3.10.

## Exclusion criteria

We restricted our analysis to conterminous US counties/county equivalents ($N = 3158$) because counties in Alaska and Hawaii have few neighboring counties and unique geological contexts impacting water contaminant concentrations. We further excluded 473 counties that did not have CWS concentration estimates available for either arsenic or uranium, 34 counties missing the percent of public drinking water supplied by groundwater sources, and 20 counties missing the percent of adults with a high school diploma. The county level CWS metal concentration estimates and sociodemographic characteristics for counties excluded from our analyses are presented in (Supplementary Table 4). After these exclusions, the total number of counties with CWS metal concentration estimates available was $N = 2585$ for arsenic and $N = 1174$ for uranium. No counties were missing racial/ethnic composition variables. We further restricted our analysis to assessing the association between county-level racial/ethnic composition and public drinking water metal concentrations for the proportion of non-Hispanic Black, American Indian/Alaskan Native, Hispanic/Latino, and non-Hispanic White residents. Analyses were conducted separately for each of the five racial/ethnic groups. We further restricted our analyses to counties with at least 100 residents in the racial/ethnic group of interest because very small population numbers could violate positivity assumptions (i.e., for some counties, the proportion of residents of a given racial/ethnic group would be zero or very near zero, and therefore effect estimates would be extrapolated beyond the range of actual values), or bias results if small increases in the absolute number of residents resulted in large percentage differences[33]. We selected 100 as the cut-point assuming most public water systems serve at least 25 households with an average of four residents per household (EPA defines public water systems as those that serve at least 15 service connections or 25 people at least 60 days per year)[79]. We did not evaluate associations for the proportion of non-Hispanic Asian and Native Hawaiian or Other Pacific Islander residents because no conterminous US counties had >60% of residents in these categories, a common cut-point to identify racial/ethnic majority areas in prior literature[80].

## Statistical analysis: descriptive data analysis

To describe sociodemographic differences and differences in public drinking water metal concentrations across all counties in our analysis versus the counties included in the analysis for each racial/ethnic group, we described mean county-level CWS arsenic and uranium, concentrations, population size, population density, the percent of public drinking water supplied from groundwater sources, median household income, the percent of residents with a high school diploma, the percent of the population living in rural areas, and racial/ethnic composition across these county categories and presented them in Table 1. We also compared these characteristics across counties that were excluded from analysis to assess potential selection bias (Supplementary Table 4).

## Statistical analysis: assessing spatial clustering

We first explored spatial autocorrelation (spatial dependence) in county-level CWS metal concentration estimates across the US. We used a simple contiguity first order queen weighting matrix with binary values identifying neighbors ($i = 1$) and non-neighbors ($i = 0$). Counties with no neighbors were excluded. We evaluated the county-level association between the five racial/ethnic composition variables and CWS mean arsenic and uranium concentrations in 8 separate ordinary least squares (OLS) linear models, regressing each CWS metal on each of the four racial/ethnic composition variables (proportion of non-Hispanic Black, American Indian/Alaskan Native, Hispanic/Latino, and non-Hispanic White residents). To assess global spatial autocorrelation, we visually assessed the residuals from each OLS model and assessed Moran's I residual scatterplots and statistics for each CWS

metal. The Moran's I statistic is the coefficient of best fit associated with the OLS line of best fit for each local unit's (i) standardized value plotted against the local unit's identified neighborhood (j) standardized average value of CWS metal concentration. To assess local spatial autocorrelation, we examined Local Indicators of Spatial Association (LISA) statistics, where every local unit's ($I_i$) LISA score is the product of the standardized units' value ($z_i$) relative to the average standard local values, multiplied by the sum of the weight matrix value times the neighborhood average value[80].

## Statistical analysis: spatial lag regression

Because OLS residuals, Moran's I scatterplots, and Moran's I statistics ($p < 0.001$) all indicated global spatial autocorrelation for arsenic and uranium, and ignoring spatial autocorrelation violates assumptions of independence and can bias effect estimates[34], we further assessed whether a spatial error or spatial lag model was most appropriate for our data using the Lagrange (Lag) Multiplier diagnostics function (lm.LMtest) in R using the package "spatialreg". A significant spatial lag or error term indicate that the residuals across spatial locations are correlated. Both the spatial lag and spatial error models returned $p$-values < 0.001, but we concluded that the spatial lag model was preferable to the spatial error model because the model estimates were larger[81]. A description of these diagnostic assessments for the OLS and spatial lag regression models are available in Supplementary Table 6.

To quantify the association between county-level racial/ethnic composition and CWS metal concentrations, we evaluated spatial lag regression models with an autoregressive correlation structure separately for CWS arsenic and uranium concentrations and for each of the five racial/ethnic composition variables in R using the lagsarlm function from the "spatialreg" package[82]. We assessed the geometric mean ratio (GMR) and corresponding percent difference of CWS arsenic and uranium concentrations per 10% higher proportion of residents classified as non-Hispanic Black, American Indian/Alaskan Native, Hispanic/Latino, and non-Hispanic White. Fully adjusted models (Model 1) were adjusted for population density, the percent of public drinking water sourced from groundwater supplies, median household income, the percent of adults with a high school diploma. Model 2 further adjusted for the proportion of the other racial/ethnic groups except for the proportion of non-Hispanic White residents. This "leave-one-out" modeling approach is useful when proportional variables sum to nearly 100% and higher proportions of one component occur simultaneously with lower proportions of at least one other component (this approach has been used in studies assessing arsenic metabolism, cell type ratios, and nutritional epidemiology)[83]. An example of the interpretation of the coefficient of interest from these models is as follows: in models assessing the proportion of Hispanic/Latino residents as the primary exposure, the coefficient for the proportion of Hispanic/Latino residents estimates the geometric mean ratio of CWS metal concentrations per 10% higher proportion of Hispanic/Latino residents because the proportion of non-Hispanic White residents is lower, controlling for the proportion of non-Hispanic Black and American Indian/Alaskan Native residents. We selected non-Hispanic White as the "leave-one-out" group because structural racism privileges and benefits this population group and we predicted that counties with lower proportions of these residents would have higher CWS metal concentrations. Because effect estimates from Model 1 and Model 2 were very similar, we selected the more parsimonious Model 1 as the main model of interest.

We performed several sensitivity analyses. First, we repeated the analyses adjusting for Model 1 covariates and the total number of non-Hispanic White residents, with similar findings (results not shown). Second, we repeated the analyses using the county-level weighted 95th percentile public drinking water metal concentrations (instead of the natural log-mean estimates), with similar findings although effect estimates were generally stronger (results not shown). Third, we

repeated the analyses by assessing the geometric mean ratio of CWS arsenic and uranium per change in the proportion of a given racial/ethnic group that corresponds to (a) a 60% higher proportion of residents, a common cut-point to identify racial/ethnic majority areas in prior literature[80], and (b) the value corresponding to the interquartile range for the proportion of residents in the specified racial/ethnic group, with similar findings for both approaches (Supplementary Table 1). Fourth, we repeated the spatial lag regression evaluating the association for higher proportions of American Indian/Alaskan Native residents without adjusting for population density because many communities and counties with a large proportion of these residents have relatively low population density (effect estimates for both arsenic and uranium were similar but stronger, results not shown). Fifth, we repeated our analyses after restricting to counties in the western US where county-level CWS arsenic and uranium estimates are highest (we defined the western US as Washington, Oregon, Idaho, Montana, Wyoming, North Dakota, South Dakota, Nebraska, Kansas, Oklahoma, Texas, New Mexico, Colorado, Utah, Arizona, Nevada, and California). Analyses restricted to western US generally yield similar findings although positive associations were identified for 10% higher proportions of non-Hispanic Black residents and uranium. However, these findings were limited by the small sample size ($n = 296$ counties; Supplementary Table 1). Finally, to complement our analyses assessing higher proportions of non-Hispanic White residents, we assessed the associations per a 10% higher proportion of all residents who were not categorized as non-Hispanic White, including non-Hispanic Black, American Indian/Alaskan Native, Hispanic/Latino, non-Hispanic Asian, Native Hawaiian or Other Pacific Islander, and all residents in other racial/ethnic categories. This sensitivity analysis is specifically intended to evaluate the prediction that higher proportions of non-Hispanic White residents is associated with lower CWS metal concentrations.

## Statistical analysis: geographically weighted regression

To explore the strength and direction of the association between county-level racial/ethnic composition and CWS arsenic and uranium concentrations and evaluate spatial non-stationarity in the effect estimates, we conducted geographically weighted regression using the "GWmodel" package in R version 4.0.3[84]. Geographically weighted regression models are exploratory, Gaussian-like, distance-based models that incorporate a spatial weighting function, combine geographic and attribute information, and obtain location-specific results. This approach provides a different regression output for each individual county, which allow us to assess variation across space in both (a) the magnitude and direction of the effect of racial/ethnic composition on CWS metal concentration, and (b) model goodness-of-fit. We ran separate geographically weighted regression models reproducing the separate models for CWS arsenic and uranium and for each of the four racial/ethnic groups that we previously assessed with global spatial lag regression models. Models were fully adjusted for population density, the percent of public drinking water sourced from groundwater supplies, median household income, and the percent of adults with a high school diploma (Model 1).

## Reporting summary

Further information on research design is available in the Nature Portfolio Reporting Summary linked to this article.

# Data availability

All data used in this analysis are publicly available. The CWS metal concentration data are available at: https://github.com/annenigra/US-PublicWaterSystem-Metal-Estimates. An interactive map of county-level CWS metal concentrations is also available at: https://msph.shinyapps.io/drinking-water-dashboard/. County-level racial/ethnic composition and county-level sociodemographic data are available at: https://www.census.gov/library/publications/2011/compendia/usa-

counties-2011.html and: https://www.countyhealthrankings.org/explore-health-rankings/rankings-data-documentation/national-data-documentation-2010-2018. The CDC Social Vulnerability Index data are available at: https://www.atsdr.cdc.gov/placeandhealth/svi/data_documentation_download.html. Data on the percent of public drinking water supplied by groundwater sources are available at: http://pubs.er.usgs.gov/publication/cir1405. A full description of all county-level variables used in this analysis is available in Supplementary Table 5. Documentation of the data sources and processed data is available in the following GitHub repository: https://github.com/annenigra/race-ethnicity-water-metals.

## Code availability

Replication code and the full dataset required to reproduce the results are maintained in the following GitHub repository[85]: https://doi.org/10.5281/zenodo.7301984.

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

## Acknowledgements

Research reported in this publication was supported by the Office Of The Director, National Institutes Of Health and the National Institute Of Dental & Craniofacial Research, under Award Number DP5OD031849 (AEN), by National Institute of Environmental Health Sciences (NIEHS) grant 2T32ES007322 (AEN), NIEHS grant P300ES009089 (AEN, ANA, BCB), NIEHS grant P42 ES033719 (AEN, ANA, BCB), and by a fellowship from La Caixa Foundation (ID100010434), fellowship code LCF/BQ/AA20/11820032 (IMM). The content is solely the responsibility of the authors and does not necessarily represent the official views of the National Institutes of Health.

## Author contributions

I.M.M.: conceptualization, methodology, formal analysis, investigation, writing (original draft), writing (review and editing), validation, and visualization. B.C.B.: methodology and writing (review and editing). O.C.B.: methodology and writing (review and editing). D.T.D: methodology and writing (review and editing). M.R.J.: methodology and writing (review and editing). K.P.P.: methodology and writing (review and editing). M.S.: methodology and writing (review and editing). S.J.P.: methodology and writing (review and editing). A.N.A.: conceptualization, methodology, and writing (review and editing). A.E.N.: conceptualization, methodology, formal analysis, investigation, writing (original draft), writing (review and editing), validation, and visualization.

## Competing interests

The authors declare no competing interests.
