## [Peer Review File · Nature Communications]

Nationwide geospatial analysis of county racial and ethnic composition and public drinking water arsenic and uraniumReviewers' Comments:

Reviewer #1:

Remarks to the Author:

In this manuscript the authors use spatial analysis methods to evaluate associations between county-level racial/ethnic composition and concentrations of arsenic and uranium. The authors use a county-level, population weighted concentration estimate for these contaminants. The study intended to evaluate global and local spatial autocorrelation among contaminant concentrations at the county scale and to investigate the association of these concentrations with racial/ethnic composition. The authors employed a spatial lag regression model and geographic weighted regression to test their hypotheses. The study results indicate strong spatial autocorrelation in CWS arsenic and uranium concentrations at the county scale. The spatial models also indicated higher arsenic and uranium concentrations in CWSs with while proportions of Hispanic/Latino and AIAN persons. Lower concentrations of these contaminants were observed as the proportion of non-Hispanic non-white persons increased. The GWR also suggests different magnitudes and directions at the regional level. The methods are appropriate and sensitivity analyses were conducted to evaluate the sensitivity of the results to various factors and subsets of the data. The study findings provide more evidence of environmental justice concerns and exposure disparities among communities of color and expand previous work to community water systems throughout the United States.

This is a strong manuscript that presents compelling information. I have a few suggestions that I believe would strengthen it.

Main Text

The authors indicate that this study is descriptive (lines 206-207, 249-251) and does not investigate the mechanisms by which disparities and environmental injustice are manufactured at the CWS and county level. The authors hypothesize that structural inequalities contribute to the observed exposure disparities but they do not test that hypothesis. Lines 71-79 seem like appropriate interpretation. Lines 97-99 and 203-207 are worded much stronger. Please review these statements to ensure the conclusion matches the scope of the study data recognizing that this is a descriptive study.

Results

Unclear what the term "positivity violations" references or means. Please review and clarify the meaning of this term. Line 142

Discussion

Lines 314-318 . Authors might also consider discussing work from the Navajo Birth Cohort study, or other aligned studies that document U exposure in Native populations in the SW.

- Dashner-Titus EJ, Hoover J, Li L, Lee JH, Du R, Liu KJ, Traber MG, Ho E, Lewis J, Hudson LG. Metal exposure and oxidative stress markers in pregnant Navajo Birth Cohort Study participants. *Free Radical Biology and Medicine*. 2018 Aug 20;124:484-92.
- Hoover JH, Erdei E, Begay D, Gonzales M, Jarrett JM, Cheng PY, Lewis J, NBCS Study Team. Exposure to uranium and co-occurring metals among pregnant Navajo women. *Environmental research*. 2020 Nov 1;190:109943.
- Murphy M, Lewis L, Sabogal RI, Bell C. Survey of unregulated drinking water sources on Navajo Nation. In *American Public Health Association 137th Annual Meeting and Exposition on Water and Public Health 2009 Nov 10 (Vol. 7, No. 11)*.

Lines 356-3645, run on sentence. Please reword for clarity.

Methods

Lines 477-493 – spatial clustering. Upon reviewing the county-level maps, what impact does the neighborhood size have on LISA statistics? Looks like some counties have zero or one neighbor while others have many more. Please review and clarify as needed.

Reviewer #2:

Remarks to the Author:

I have read with interest the manuscript entitled "Nationwide geospatial analysis of country-level racial/ethnic composition and public drinking water arsenic and uranium". This is an interesting (and new, to my knowledge) approach linking public water arsenic and uranium and racial/ethnic composition and is important in the context of environmental inequalities. I have some major (and some more minor) concerns however that need to be clarified.

Firstly, there is an inverse relationship between geogenic arsenic and uranium that is well established due to the differing redox controls on mobility. It is unclear if/how the dataset used here follows that well-established geochemical relationship. Indeed the statements about non-Hispanic White residents associated with lower arsenic and uranium (or non Hispanic Black residents associated with higher arsenic and uranium) does not seem to be consistent with the generally inverse relationship between arsenic and uranium. This needs to be addressed – in the first instance a direct plot of mean (or better median) As versus mean U (at county level?) would be a beneficial addition as well as appropriate discussion on this relationship.

Secondly, much of the discussion is very qualitative. Most of the numbers in Results are descriptive and simply reporting demographic data, with less focus on the specific relationships between racial/ethnic composition and arsenic/uranium. For example – when statements are made about certain populations having lower arsenic or uranium – specifically how much lower? Are the differences statistically significant? When cities or regions are mentioned, this is often rather vague – can specific cities/zones/etc be identified even as examples where various scenarios are happening?

Thirdly, the "Results" section seems very focussed on methods. It would be improved if the Results section focussed more specifically on key and specific findings (backed up with specific quantified values) and then relevant discussion.

More minor comments – it would be beneficial if the color scale on Figure 1 (and others where relevant) reflected the guideline values for As and U.

Language throughout needs to be checked for accuracy – for example line 64 refers to a standard in the Netherlands for 1 ug/L – my understanding based on the cited paper Ahmad et al 2020 is that the water companies are targeting a level of 1 ug/L but that is NOT a formal standard.

REVIEWER COMMENTS

Reviewer #1:

In this manuscript the authors use spatial analysis methods to evaluate associations between county-level racial/ethnic composition and concentrations of arsenic and uranium. The authors use a county-level, population weighted concentration estimate for these contaminants. The study intended to evaluate global and local spatial autocorrelation among contaminant concentrations at the county scale and to investigate the association of these concentrations with racial/ethnic composition. The authors employed a spatial lag regression model and geographic weighted regression to test their hypotheses. The study results indicate strong spatial autocorrelation in CWS arsenic and uranium concentrations at the county scale. The spatial models also indicated higher arsenic and uranium concentrations in CWSs with while proportions of Hispanic/Latino and AIAN persons. Lower concentrations of these contaminants were observed as the proportion of non-Hispanic non-white persons increased. The GWR also suggests different magnitudes and directions at the regional level. The methods are appropriate and sensitivity analyses were conducted to evaluate the sensitivity of the results to various factors and subsets of the data. The study findings provide more evidence of environmental justice concerns and exposure disparities among communities of color and expand previous work to community water systems throughout the United States.

This is a strong manuscript that presents compelling information. I have a few suggestions that I believe would strengthen it.

Main Text

The authors indicate that this study is descriptive (lines 206-207, 249-251) and does not investigate the mechanisms by which disparities and environmental injustice are manufactured at the CWS and county level. The authors hypothesize that structural inequalities contribute to the observed exposure disparities but they do not test that hypothesis. Lines 71-79 seem like appropriate interpretation. Lines 97-99 and 203-207 are worded much stronger. Please review these statements to ensure the conclusion matches the scope of the study data recognizing that this is a descriptive study.

Response: We thank the reviewer for these comments, and agree that the description of our hypothesis and testable prediction were not clear as written. Prior individual studies have identified specific mechanisms (including those related to structural racism) by which racial/ethnic inequalities in water exposures are created and reinforced- these studies provide the basis for the leading theoretical framework in this area (the "Drinking Water Disparities Framework", Balazs et al., 2014, *Am J Public Health*, PMID: PMC4025716). The reviewer is correct that our testable prediction is related to the association between racial/ethnic composition and water metal concentrations, not mechanisms responsible for these inequities. Based on the Drinking Water Disparities Framework and existing literature, we hypothesized that any inequities we found in water contaminant concentrations by racial/ethnic composition were related to White supremacy and structural racism (which is not directly testable in this study but is supported by our complementary analysis which isolates the impact of the absence of non-Hispanic White residents). We have now revised these statements and included supporting references:

Pages 3-4, lines 122-130: " We predicted that higher proportions of non-Hispanic White residents would be associated with lower CWS arsenic and uranium concentrations (likely reflecting White supremacy and structural racism related to public drinking water infrastructure, investment decisions, and regulatory action that specifically benefit non-Hispanic White communities, which have been previously documented for some water exposures)^{23,24,26,27}, and that higher proportions of Hispanic/Latino and American Indian residents would be associated with higher CWS arsenic and uranium concentrations

(also consistent with previous documented inequalities in public drinking water infrastructure and regulatory action across population groups)^{20,23,28,29}.

For this sentence (lines 203-207 in the previous draft, lines 287-294 in the current draft), we are specifically interpreting the finding of our complementary analysis which combines all residents not categorized as non-Hispanic White into one group (allowing us to specifically isolate the impact of the absence of non-Hispanic White residents, or alternatively, the potential protective impact of non-Hispanic White residents):

Page 8, lines 281-288: “Because this association persisted even after combining all residents not categorized as non-Hispanic White, these complementary findings support the general hypothesis and leading theoretical framework on water exposure disparities—that White supremacy and structural racism are associated with drinking water inequalities²³. However, the specific mechanisms underlying these disparities (e.g., infrastructure investments or management decisions specifically benefiting non-Hispanic White communities, etc.) are beyond the scope of this analysis. Findings were also similar for barium and selenium (Supplementary Table 3).”

Results

Unclear what the term “positivity violations” references or means. Please review and clarify the meaning of this term. Line 142

Response: We now clarify the meaning of “positivity” and “positivity violations” and the relevance of these terms to our work and included explanatory references:

Page 5, lines 183-187 (results section): “Because we restricted our analysis of each racial/ethnic group to counties with at least 100 residents of that group to avoid positivity violations³³ (i.e., the number of residents in a given racial/ethnic subgroup should be greater than zero for all counties analyzed or effect estimates would be extrapolated beyond the range of actual values), the number of counties included in the analysis of each racial/ethnic group differed (see Methods: Exclusion criteria).”

Pages 17-18, lines 580-588 (methods section): “We further restricted our analyses to counties with at least 100 residents in the racial/ethnic group of interest because very small population numbers could violate positivity assumptions (i.e., for some counties, the proportion of residents of a given racial/ethnic group would be zero or very near zero, and therefore effect estimates would be extrapolated beyond the range of actual values), or bias results if small increases in the absolute number of residents resulted in large percentage differences³³.”

Discussion

Lines 314-318 . Authors might also consider discussing work from the Navajo Birth Cohort study, or other aligned studies that document U exposure in Native populations in the SW.

- Dashner-Titus EJ, Hoover J, Li L, Lee JH, Du R, Liu KJ, Traber MG, Ho E, Lewis J, Hudson LG. Metal exposure and oxidative stress markers in pregnant Navajo Birth Cohort Study participants. *Free Radical Biology and Medicine*. 2018 Aug 20;124:484-92.
- Hoover JH, Erdei E, Begay D, Gonzales M, Jarrett JM, Cheng PY, Lewis J, NBCS Study Team. Exposure to uranium and co-occurring metals among pregnant Navajo women. *Environmental research*. 2020 Nov 1;190:109943.
- Murphy M, Lewis L, Sabogal RI, Bell C. Survey of unregulated drinking water sources on Navajo Nation. In *American Public Health Association 137th Annual Meeting and Exposition on Water and Public Health 2009 Nov 10 (Vol. 7, No. 11)*.

Response: We thank the reviewer for these valuable references and now include several in our discussion section. We cite the Dashner-Titus et al. manuscript in discussion of arsenic and uranium associated health outcomes, including sub-clinical endpoints such as oxidative stress.

Page 12, lines 408-412 “. At moderate to high levels of exposure, inorganic arsenic is associated with a number of adverse health outcomes including diabetes, lung disease, neurologic effects, kidney disease, and birth outcomes including low birth weight and impaired neurodevelopment; inorganic arsenic is also associated with cardiovascular disease and increased oxidative stress even at low to moderate exposure levels common in the US population ^{1,5,49-52}. “

We included the Hoover J.H. et al., manuscript as a supporting example of the high burden of exposure to uranium in the American Indian Reservations.

Pages 12-13, lines 422-433: “For example, the mean uranium concentration in urine (biomarker of internal dose) among pregnant women living in the American Indian Reservations in the Southwest is 2.67 to 2.8 times higher compared to the general US population⁵⁵. ”

We included the Murphy et al., manuscript when discussing the high levels of arsenic and uranium in drinking water in the Navajo Nation, as well as to address a different reviewer comment regarding the co-occurrence of arsenic and uranium in drinking water sources.

Page 12, lines 418-422: “For example, American Indian/Alaskan Native populations experience substantially higher rates of kidney cancer compared to non-Hispanic White populations, especially in the Southwestern US where uranium exposures and co-exposures to arsenic and other metal mixtures are relatively high as a result of the uranium mining activity that occurred over the past century^{19,44,45,54}”

Page 12, lines 401-402: “The co-occurrence of arsenic and uranium has been documented in both regulated and unregulated water supplies ^{16,19,32,44,45}”

Lines 356-3645, run on sentence. Please reword for clarity.

Response: Thank you, we have revised this:

Page 14, lines 476-481: “Geographically weighted regression indicated a positive association between larger proportions of non-Hispanic Black residents and CWS arsenic in parts of west Texas and the Midwest. These findings suggest that additional analyses at a finer geographic resolution should focus specifically on these particularly vulnerable geographic areas where metals in CWSs and soil are particularly high (public water system distribution boundaries are also publicly available for many of these states).”

Methods

Lines 477-493 – spatial clustering. Upon reviewing the county-level maps, what impact does the neighborhood size have on LISA statistics? Looks like some counties have zero or one neighbor while others have many more. Please review and clarify as needed.

Response: The reviewer is correct that spatial lag models depend on neighbors and their size. To address this concern. Counties without neighbors (and all counties in AK and HI which we excluded because of their unique geologic context) are not included, as spatial lag modeling cannot accommodate spatial units with no neighbors (necessary to compute the spatial weight matrix). Differences in county size across the US may result in larger measurement error in areas with larger county area sizes (e.g., the western US). That said, counties that are large do not necessarily have more neighbors because they tend to be found adjacent to other larger counties (See Fig 1 in the manuscript). As such, the lag model generally works for most counties. It is however, important to recognize that the spatial scale of water quality variance is frequently quite high (typically less than a km, Podgorski et al, 2020, *Science*, PMID: 32439786,

Ayotte et al, 2015, *Sci Total Environ*. PMID: 24650751, Connolly et al., 2022, *Environ Sci Technol*, PMID: 34951307), meaning that most counties (indeed most political units of any sort, even towns and cities) contain highly variable water quality. The maps generated using spatial lag models thus show regional trends at the spatial scale of the lag, but cannot capture this fine-scaled variance. We now clarify how differences in county size may impact our interpretation:

Page 13, lines 441-445: “Future studies at smaller spatial scales and with finer geographic resolution may also address the substantial differences in county size across the US. Larger county sizes may lead to larger measurement error in water metal concentrations, which may especially impact findings in the western US compared to the eastern US where counties areas are smaller.”

Reviewer #2:

I have read with interest the manuscript entitled “Nationwide geospatial analysis of country-level racial/ethnic composition and public drinking water arsenic and uranium”. This is an interesting (and new, to my knowledge) approach linking public water arsenic and uranium and racial/ethnic composition and is important in the context of environmental inequalities. I have some major (and some more minor) concerns however that need to be clarified.

Firstly, there is an inverse relationship between geogenic arsenic and uranium that is well established due to the differing redox controls on mobility. It is unclear if/how the dataset used here follows that well-established geochemical relationship. Indeed the statements about non-Hispanic White residents associated with lower arsenic and uranium (or non Hispanic Black residents associated with higher arsenic and uranium) does not seem to be consistent with the generally inverse relationship between arsenic and uranium. This needs to be addressed – in the first instance a direct plot of mean (or better median) As versus mean U (at county level?) would be a beneficial addition as well as appropriate discussion on this relationship.

We thank the reviewer for this comment, in particular because it highlights one of the significant findings of this work, that both As and U are often high in the same communities. Our bias going into this work would have similarly been that these contaminants are generally inversely correlated due to the fact that As is usually associated with reducing environments (e.g., Fendorf et al, 2010, *Science*. PMID: 20508123), while U is most soluble and mobile in oxidizing, alkaline environments where the uranyl cation forms carbonate complexes. This work finds that this assumption is not as true as we might have expected, particularly at levels closer to the MCL. While we agree that the co-occurrence of arsenic and uranium was previously unexpected, several recent studies also have documented that arsenic and uranium do co-occur in both regulated and unregulated water supplies at both the regional and national level in the US (including one paper suggested by Reviewer 1).

While Reviewer 1 notes that the analysis in this manuscript does not explain why contamination occurs, our previous research and the results in this manuscript provides insight into the mechanisms of release that help explain this correlation. We recently published an analysis illustrating the strong clustering of uranium, arsenic, and selenium in regulated community water systems across the US using the water system level-version of the database used in this study (Ravalli et al. 2022, *Lancet Planetary Health*, PMID: PMC9037820). Similar correlations were seen in some wells within a smaller area in South Dakota (Spaur et al., 2020, *Sci. Tot. Environ*. PMID: 33991916). These publications attributed the common occurrence of As and U in waters to the formation of carbonate complexes may be particularly relevant for increasing solubility, through aqueous complex formation and by stimulating desorption from mineral surfaces. This Ravalli paper presents a correlation and hierarchical cluster analysis describing the clustering of uranium and arsenic in these community water systems, we did not repeat that analysis or a similar one at the county-level in this manuscript. Instead, this manuscript focuses on the novel aspects of the association between drinking water quality in public supplies, and race/ethnicity.

While we agree with Reviewer 1 that this analysis, which does not include other elements or water quality parameters needed to appropriately discuss mechanisms that explain the observed trends, we do think it is appropriate to use previous research and the similar spatial correlations with race to contextualize our results with a plausible mechanism that explains them. To do so, we have updated the manuscript to include a brief description of this co-occurrence and several supporting references in the Discussion:

Page 12, lines 399-405: “Across all analyses, our findings were largely consistent for arsenic and uranium, likely reflecting the clustering and co-occurrence of arsenic and uranium in public water systems which has been previously described in detail¹⁶. The co-occurrence of arsenic and uranium has been documented in both regulated and unregulated water supplies^{16,19,32,44,45}, and may be related to carbonate complexes increasing the solubility of arsenic and uranium, and/or the presence of bicarbonate ions in oxalic water sources (bicarbonate ions may be especially relevant for the continued co-occurrence of arsenic and uranium in treated water)⁴⁶⁻⁴⁸.”

We also modified Figure 1 to provide a better visualization of the range of arsenic and uranium concentrations across the US, as suggested in a following reviewer comment.

Secondly, much of the discussion is very qualitative. Most of the numbers in Results are descriptive and simply reporting demographic data, with less focus on the specific relationships between racial/ethnic composition and arsenic/uranium. For example – when statements are made about certain populations having lower arsenic or uranium – specifically how much lower? Are the differences statistically significant? When cities or regions are mentioned, this is often rather vague – can specific cities/zones/etc be identified even as examples where various scenarios are happening.

Response: We thank the reviewer for this comment, and we have now revised this section as follows:

We named the relevant states/areas where LISA clusters were identified:

Page 5, lines 173-178: “Areas with High-High significant spatial autocorrelation for both arsenic and uranium were located in the Midwest and Southwest (e.g., Nevada, California, Arizona, New Mexico, Texas, Kansas, Nebraska) where CWS arsenic and uranium concentrations are highest in underlying groundwater^{16,17,32}. Low-Low clusters for arsenic were identified in the Southeast and Mid-Atlantic regions (e.g., Alabama, Arkansas, North and South Carolina, Virginia, Kentucky).”

We include mean water metal concentrations when describing Table 1:

Page 6, lines 216-222: “Compared to counties included in any analysis, counties included in the analysis for non-Hispanic Black residents had lower mean CWS arsenic (1.17 µg/L vs. 1.42µg/L) and uranium (2.5 µg/L vs. 3.45 µg/L) concentrations, a lower percentage of CWSs reliant on groundwater, and a lower percentage of the population living in rural areas. Counties included in the analysis for American Indian/Alaskan Native residents had larger population size and density and the lowest percentage of the population living in rural areas. Counties included in the analysis for Hispanic/Latino residents had higher mean CWS uranium (3.38 µg/L vs. 3.45 µg/L) concentrations”

We did not substantially edit the section describing the association between racial/ethnic composition and water metal concentrations in global spatial models, as these results were already described quantitatively (e.g., “In fully adjusted models (Model 1), the percent increase in the geometric mean (95% CI) of CWS metal concentrations per 10% higher proportion of Hispanic/Latino residents was 6% (4, 8) for arsenic and 17% (13, 22) for uranium.”). Although the percent change in the geometric mean water metals was derived from the geometric mean ratios (GMRs), we only present the percent change findings in the Results section because the

percent change is the main effect estimate of interest (GMRs are presented in Table 2 for interested readers).

We now clarify how we assessed statistical significance:

Pages 7, lines 250-251: “Effect estimates were considered statistically significant when 95% confidence intervals did not cross the null (GMR of 1 and percent change of 0).”

For brevity and clarity, we also only discuss results from sensitivity analyses qualitatively in the Results section, but we do make quantitative findings available in the Supplemental material.

When reporting the results from the geographically weighted regression (GWR), we now include state names. Because GWR outputs a specific effect estimate for each county, it is not typical or concise to quantitatively report effect estimates for individual counties. Rather, it is typical to describe general spatial patterns in county-specific effect estimates and how they relate to findings from global spatial models:

Page 9, lines 323-330: “This was particularly evident for arsenic in western Texas, Arizona, California, Nevada, Utah, Oregon and Washington, and for uranium in Arizona, California, Nevada, Oregon and Oklahoma where we observed positive associations of higher magnitude (Figures 3 and 4). Similarly, we observed a positive association between higher proportions of American Indian/Alaskan Native residents and CWS arsenic and uranium in the Northern Plains and in the west, particularly across Oregon, Washington, Montana, North Dakota and South Dakota, Minnesota, and Wisconsin for arsenic, and California, Nevada, Oregon, North Dakota, and Wisconsin for uranium.”

We are unable to report or interpret our findings at the city level because our data are at the county level:

Page 13, lines 435-445: “First, our nationwide evaluation of the county-level association between racial/ethnic composition and CWS metal concentrations may mask associations operating at finer geographic resolutions (e.g. within states, or across Census blocks or zip-codes)^{21,56}. We were not able to evaluate these associations at finer geographic resolutions because nationwide CWS concentration estimates are not yet available at these levels of aggregation (there is currently no publicly available, nationwide map of public drinking water system distribution boundaries that can be used to generate concentration estimates at these resolutions). Future studies at smaller spatial scales and with finer geographic resolution may also address the substantial differences in county size across the US. Larger county sizes may lead to larger measurement error, which may especially impact findings in the western US compared to the eastern US where counties areas are smaller.”

Thirdly, the “Results” section seems very focussed on methods. It would be improved if the Results section focussed more specifically on key and specific findings (backed up with specific quantified values) and then relevant discussion.

Response: We agree that the Results section provided some background to the methods. We have now revised and edited the Results section to reduce the description of the methods and focus more on quantitative findings (as described in response to the previous comment). However, because the methods we use are field-specific, the journal readership is a general scientific audience, and the Results section is presented before the Methods section, we kept a few contextualizing sentences in the Results section and keep our references to the detailed Methods section areas. We hope this improves the flow of the paper for a general audience that may be not familiar with spatial environmental epidemiology.

We removed the following details from the Results section (these are covered in the Methods section):

- Details on the Moran’s I statistic (previously on Page 5)

- Details on spatial clustering and model choice (previously on Page 6)
- Utility of GWR (previously on Page 8)

We can make additional edits to the reorganization of this section if preferred by the editor.

More minor comments – it would be beneficial if the color scale on Figure 1 (and others where relevant) reflected the guideline values for As and U.

Response: Thank you for this suggestion, we have now modified Figure 1 to include the EPA MCL values for arsenic and uranium as suggested.

Language throughout needs to be checked for accuracy – for example line 64 refers to a standard in the Netherlands for 1 ug/L – my understanding based on the cited paper Ahmad et al 2020 is that the water companies are targeting a level of 1 ug/L but that is NOT a formal standard.

Response: We have now clarified this sentence, noting that the guideline in the Netherlands was voluntarily adopted by the Dutch Association of Drinking Water Companies. In addition to the originally cited publication (Ahmad et al 2020) we also now include an additional reference: *Integrating arsenic in water safety planning in The Netherlands. Environmental Arsenic in a Changing World* 618–619 (CRC Press, 2019). doi:10.1201/9781351046633-244.

Page 2, lines 70-75: “Although the US EPA sets a maximum contaminant level (MCL) of 30 µg/L for uranium and 10 µg/L for arsenic, EPA’s non-enforceable maximum contaminant level goal for both is 0 µg/L because there is no known safe level of exposure to either arsenic or uranium (Denmark and the states of New Jersey and New Hampshire set more health-protective arsenic MCLs of 5 µg/L, and drinking water providers in the Netherlands adopted a voluntary standard of 1 µg/L in 2016)^{12–15}.”

Reviewers' Comments:

Reviewer #1:

Remarks to the Author:

This manuscript is a revised version of a previously submitted manuscript titled, "Nationwide geospatial analysis of county-level racial/ethnic composition and public drinking water arsenic and uranium." The manuscript reads well and the authors directly addressed my previous concerns. Namely, clarifying the hypothesis driven-nature of this work and ensuring the interpretation matched the scope of analysis. In my opinion the authors have fully responded to my questions. The results are compelling, the work is significant, and the methodology is sound.

Please review the manuscript once more for typos (example Line 402 carbonato instead of carbonate). Otherwise I have no further comments or questions.

Reviewer #2:

Remarks to the Author:

The authors have provided a considered and thorough reply to both reviewers' comments. The changes that have been made to address the reviewers' comments have substantially improved the manuscript. Most of their rebuttal points are well-made, and upon re-review I have only a few minor remaining comments.

[1] I still think it would be useful to specifically mention the statistical significance of the differences of the population means – e.g. are the differences of mean values of 3.38 vs 3.45 ug/L for U statistically significant? I would suggest adding some further info (e.g. p values) in the relevant main text to help support the statements made around Table 1.

[2] Line 245 – please supplement with some references on the “growing body of literature” on inequalities regarding DW contaminant distribution

[3] Same as line 257 – 258 – needs some supporting references

[4] Line 306 – typo – carbonate?

[5] Line 356 – very few counties had $N > 100$ people for Asian groups? Is this correct? This is surprising to me.

Thanks again to the authors for their thorough revision.

We would like to thank both reviewers for their very helpful comments and feedback. We have revised the manuscript following their constructive feedback and have also edited the format of the manuscript to comply with the journal requirements and we ensured compliance with the data availability guidelines.

All line numbers referenced here correspond to the modified document as the clean version has different line numbers. All changes have been highlighted in the text using yellow color.

REVIEWER COMMENTS

Reviewer #1 (Remarks to the Author):

This manuscript is a revised version of a previously submitted manuscript titled, "Nationwide geospatial analysis of county-level racial/ethnic composition and public drinking water arsenic and uranium." The manuscript reads well and the authors directly addressed my previous concerns. Namely, clarifying the hypothesis driven-nature of this work and ensuring the interpretation matched the scope of analysis. In my opinion the authors have fully responded to my questions. The results are compelling, the work is significant, and the methodology is sound.

Please review the manuscript once more for typos (example Line 402 carbonato instead of carbonate). Otherwise I have no further comments or questions.

Response: Although both “carbonate” and “carbonato” are appropriate, we have changed this to “carbonate” as it is more recognizable.

Response:

Reviewer #2 (Remarks to the Author):

The authors have provided a considered and thorough reply to both reviewers' comments. The changes that have been made to address the reviewers' comments have substantially improved the manuscript. Most of their rebuttal points are well-made, and upon re-review I have only a few minor remaining comments.

[1] I still think it would be useful to specifically mention the statistical significance of the differences of the population means – e.g. are the differences of mean values of 3.38 vs 3.45 ug/L for U statistically significant? I would suggest adding some further info (e.g. p values) in the relevant main text to help support the statements made around Table 1.

Response: Table 1 describes the CWS arsenic and uranium concentrations in the counties included in analyses for each racial/ethnic group, and is not a comparison of water metal concentrations across population groups (See methods: “we restricted our analyses to counties with at least 100 residents in the racial/ethnic group of interest to avoid positivity violations”). Because Table 1 is a descriptive table and there is significant overlap in counties across columns, the use of inferential statistics would not be useful to inform the manuscript analyses (see *Murphy, K. (2021). In praise of Table 1: The importance of making better use of descriptive statistics. Industrial and Organizational Psychology, 14(4), 461-477. doi:10.1017/iop.2021.90.*

We have modified the text in the results and the methods sections to clarify the descriptive nature of the Table 1 presented in the manuscript.

Page 5, lines 144-148: Across the counties included in each racial/ethnic analysis and all counties included in any analysis (N= 2,653), we first described mean county-level CWS arsenic and uranium concentrations, racial/ethnic composition, and other county-level characteristics of interest (Table 1).

Page 18, lines 505-512: To describe sociodemographic differences and differences in public drinking water metal concentrations across all counties in our analysis versus the counties included in the analysis for each racial/ethnic group, we described mean county-level CWS arsenic and uranium, concentrations, population size, population density, the percent of public drinking water supplied from groundwater sources, median household income, the percent of residents with a high school diploma, the percent of the population living in rural areas, and racial/ethnic composition across these county categories and presented them in Table 1.

[2] Line 245 – please supplement with some references on the “growing body of literature” on inequalities regarding DW contaminant distribution

Response: Relevant references to support the statement have been added.

[3] Same as line 257 – 258 – needs some supporting references

Response: Relevant references to support the statement have been added.

[4] Line 306 – typo – carbonate?

Response: Although both “carbonate” and “carbonato” are appropriate, we have changed this to “carbonate” as it is more recognizable.

[5] Line 356 – very few counties had N > 100 people for Asian groups? Is this correct?

This is surprising to me.

Response: We did not analyze associations for racial/ethnic groups when no counties had at least 60% of residents of that group. We further restricted the analyses for individual racial/ethnic groups to counties with N >100 people to avoid positivity violations. We have now clarified our criteria in the manuscript.

Page 14, lines 354-356: “We did not assess associations for Asian and Native Hawaiian or Pacific Islander groups because no counties had >60% of residents for these groups.”

Pages 18, lines 499-502: We did not evaluate associations for the proportion of non-Hispanic Asian and Native Hawaiian or Other Pacific Islander residents because no conterminous US counties had >60% of residents in these categories, a common cut-point to identify racial/ethnic majority areas in prior literature ⁶⁶.

Thanks again to the authors for their thorough revision.

##

Additional comments extracted from the Author Checklist:

The manuscript contains three brief references to the role of white supremacy privilege in potentially driving the results observed (Lines 124, 284, and 355). While these statements may be true, the analysis in this manuscript does not look specifically at social drivers that may explain these findings. We ask that you edit these statements to reflect the analysis carried out in the manuscript. Alternatively, you may be able to address the relationship/implications of the

findings to broader social realities like structural racism or white supremacy in a more detailed way in the discussion section.

Response: We have extended our discussion to address the relationship of our findings to broader social realities like structural racism or white supremacy in a more detailed way, including additional supporting references in order to comply with the additional revision provided in the Author Checklist.

Pages 10-11, lines 259-280: Structural racism refers to the “totality of ways in which societies foster racial discrimination, through mutually reinforcing inequitable systems (in housing, education, employment, earnings, benefits, credit, media, health care, criminal justice, and so on).”^{36,37} Structural racism serves to maintain white supremacy, in part, by amassing resources for and benefiting white communities to the detriment of communities of color³⁸⁻⁴⁰. Extensive research in the fields of sociology and epidemiology support that racism is a fundamental cause of disease and producer of health inequalities across racial/ethnic groups.⁴¹⁻⁴³ Structural racism produces health inequalities for marginalized communities through numerous mechanisms, including through the creation and perpetuation of inequalities in toxic environmental exposures, social determinants of health, and other psychosocial stressors.^{36,37,39,44-46} Prior studies identifying the specific mechanisms underlying water inequalities are abundant, although responsible mechanisms differ across contaminants, geographic regions, and the sociodemographic characteristics of impacted communities. Briefly, previous studies have identified community linguistic isolation, selective enforcement of drinking water regulations, the underbounding of communities of color from municipal boundaries and water services, and the direct withholding of resources and infrastructure investments as examples of mechanisms responsible for producing and maintaining inequalities in exposure to regulated drinking water contaminants.^{23,26,47-49} For example, inequalities in public water lead exposure are related to changes in water supply sources, corrosion control, service line infrastructure investments, and targeted tap sampling for compliance monitoring^{32,50-53}. Similar mechanisms may underlie the unequal distribution of arsenic and uranium concentrations in CWS observed in this study, although additional research is needed to evaluate those mechanisms in detail.